# Evaluating deep learning-based melanoma classification using immunohistochemistry and routine histology: A three center study

**Christoph Wies**[1,2☯], **Lucas Schneider**[1☯], **Sarah Haggenmüller**[1], **Tabea-Clara Bucher**[1], **Sarah Hobelsberger**[3], **Markus V. Heppt**[4], **Gerardo Ferrara**[5], **Eva I. Krieghoff-Henning**[1‡], **Titus J. Brinker**[1‡]*

1 Digital Biomarkers for Oncology Group, German Cancer Research Center (DKFZ), Heidelberg, Germany, 2 Medical Faculty, University Heidelberg, Heidelberg, Germany, 3 Department of Dermatology, Faculty of Medicine and University Hospital Carl Gustav Carus, Technische Universität Dresden, Dresden, Germany, 4 Department of Dermatology, Uniklinikum Erlangen, Friedrich-Alexander-Universität Erlangen-Nürnberg, Erlangen, Germany, 5 Anatomic Pathology and Cytopathology Unit—Istituto Nazionale Tumori di Napoli, IRCCS "G. Pascale", Naples, Italy

☯ These authors contributed equally to this work.
‡ EIKH and TJB also contributed equally to this work.
* titus.brinker@dkfz.de

**Data Availability Statement:** All relevant data are within the manuscript and its Supporting Information files.

## Abstract

Pathologists routinely use immunohistochemical (IHC)-stained tissue slides against MelanA in addition to hematoxylin and eosin (H&E)-stained slides to improve their accuracy in diagnosing melanomas. The use of diagnostic Deep Learning (DL)-based support systems for automated examination of tissue morphology and cellular composition has been well studied in standard H&E-stained tissue slides. In contrast, there are few studies that analyze IHC slides using DL. Therefore, we investigated the separate and joint performance of ResNets trained on MelanA and corresponding H&E-stained slides. The MelanA classifier achieved an area under receiver operating characteristics curve (AUROC) of 0.82 and 0.74 on out of distribution (OOD)-datasets, similar to the H&E-based benchmark classification of 0.81 and 0.75, respectively. A combined classifier using MelanA and H&E achieved AUROCs of 0.85 and 0.81 on the OOD datasets. DL MelanA-based assistance systems show the same performance as the benchmark H&E classification and may be improved by multi stain classification to assist pathologists in their clinical routine.

## Introduction

Melanoma diagnoses have increased in recent decades [1] and melanoma is the fifth most common cancer in the United States [2]. In spite of its relatively high frequency, melanoma is often difficult to be histopathologically differentiated from nevi, a high diagnostic discordance rate having been reported even among experienced histopathologists [3]. If a melanoma is initially misclassified as nevus and therefore diagnosed at a later stage, the patient's chances of survival could be significantly reduced and therapy will probably have to be more intense. On

**Funding:** The presented work was funded by the federal Ministry of Health, Berlin, Germany (grants: Tumor Behavior Prediction Initiative (TPI) and Skin Classification Project 2 (SCP2)); Ministry of Social Affairs, Health and Integration of the Federal State Baden-Württemberg, Germany (grant: KTI); grant holder in all cases: Titus J. Brinker, German Cancer Research Center, Heidelberg, Germany). The sponsors had no role in the design and conduct of the study; collection, management, analysis, and interpretation of the data; preparation, review, or approval of the manuscript; and decision to submit the manuscript for publication.

**Competing interests:** TJB would like to disclose that he is the owner of Smart Health Heidelberg GmbH (Handschuhsheimer Landstr. 9/1, 69120 Heidelberg, Germany) which develops mobile apps, outside of the submitted work. SHo reports clinical trial support from Almirall and speaker's honoraria from Almirall, UCB and AbbVie and has received travel support from the following companies: UCB, Janssen Cilag, Almirall, Novartis, Lilly, LEO Pharma and AbbVie outside the submitted work. This does not alter our adherence to PLOS ONE policies on sharing data and materials.

**Abbreviations:** AI, Artificial intelligence; AUROC, Area under receiver operating characteristics curve; CI, Confidence Interval; CNN, Convolutional neural network; DL, Deep Learning; H&E, Hematoxylin and Eosin; IHC, Immunohistochemical; InD, Internal distribution; MART1, Melanoma antigen recognized by T-cells 1; MelanA, Melanocytic antigen; OOD, Out of distribution; WSI, Whole slide image.

the other hand, if harmless benign lesions are diagnosed as melanoma, the patient will suffer an unnecessary psychological and physical burden. In individual cases, overdiagnosis can even lead to unnecessary, expensive and stressful therapies, which can also be associated with high costs in the healthcare system and unnecessary toxicity for affected patients [4]. More precise diagnostic options could contribute to overcoming these problems.

Due to rapid technological advances of the last few years, AI-based assistance systems may become powerful tools for pathological cancer diagnostics. Deep Learning (DL) with Convolutional Neural Networks (CNN) has shown promise in studies aimed at distinguishing melanomas and nevi on digitized hematoxylin and eosin (H&E)-stained whole slide images (WSI) [5, 6]. In some cases, the DL approach could even outperform humans [7]. However, accuracy of these classifiers especially on external data still shows room for improvement.

In addition to standard H&E-stained slides, immunohistochemical (IHC)-stained tissue sections are often available for many cancer entities and represent a source of complementary prognostic and/or predictive information in addition to H&E-stained tissue. However, the analysis of IHC-stained slides by DL models is a relatively new area of research. Recent studies, however, have employed DL for successful classification of non-skin cancer entities, i.e., to determine HER2 status in breast cancer [8] and immune cell multistains as prognostic and predictive biomarkers in colorectal cancer [9] on IHC-stained slides. Moreover, as shown in previous work, the fusion of different data modalities often improves generalizability and performance of DL models [9–11].

IHC-stains routinely used by pathologists to better differentiate other, usually benign lesions from melanomas are MelanA (MART-1) [12], HMB-45 [13], Ki-67 [14], tyrosinase [15], S100 [16] and PRAME [17]. The expression of melanocyte antigen MelanA (also called melanoma antigen recognized by T-cells 1 (MART1)) is a lineage-specific melanocytic marker which is commonly used by histopathologists for routine diagnosis of melanocytic neoplasms, since it highlights the cytomorphology and the distribution of melanocytes.

IHC expression of MelanA can be automatically analyzed using state-of-the-art artificial intelligence (AI) methods. In this study, we investigate the use of DL-based image analysis models on MelanA-stained tissue for melanoma classification in comparison and in addition to the standard H&E-based diagnosis.

## Materials and methods

The presented study investigates melanoma suspicious lesions based on dermatoscopic investigation, which were verified histopathologically as melanoma or nevus. We use DL models to classify whether a lesion is a melanoma or a nevus based on MelanA or H&E stained tumor tissue or a combination of both stains.

Ethics approval was obtained from the ethics committees of the technical university in Dresden. Patients provided informed written consent. This work was performed in accordance with the Declaration of Helsinki [18].

### Datasets

The inclusion criteria to participate in our study was to be 18 years old with melanoma-suspicious skin lesions that were biopsied after dermoscopic examination. Suspicious lesions that were pre-biopsied or located near the eye, under the fingernails or toenails were excluded. The ground truth labels were histopathological confirmed by at least one reference dermatopathologist investigating at least the H&E-stained reference slide. MelanA (MART-1) [19, 20] immunohistochemical (IHC) and Hematoxylin and Eosin (H&E) stained tissue slides from the university hospital in Dresden were used for training, validation and hold-out testing. Slides

from the university hospital in Erlangen and from the National Cancer Institute of Naples were used for out of distribution (OOD) testing. Table 1 describes the population of all three cohorts. The Dresden, Erlangen and Naples cohorts were collected prospectively, with data collection starting in April 2021 and ending in April 2023, participants provided informed written consent. Data was physically transferred in batches. Data received before 2023 from the university hospital in Dresden was used as a training set, data received later was used as a holdout test dataset. The labels of the datasets were pathologically verified. All 3 cohorts differ in the stains of MelanA slides. Antibodies from different manufacturers with different dilutions were used at each site (Table 2). Representative WSI thumbnails from all 3 cohorts and for both stains are shown in the supplements in S1 Fig.

## Pre-processing

IHC and adjacent H&E slides from the Dresden, Erlangen and Naples cohorts were digitized with an Aperio® AT2 Slide Scanner with a 40× magnification resulting in WSIs with a resolution of 0.25 μm/px. Tumor boundaries were manually annotated under expert supervision with the QuPath digital pathology software version 0.3 [21]. WSIs were tessellated into patches of 237 px x 237 px by an in-house developed QuPath script for each slide in different (40x, 20x, 10x, 5x) magnifications for IHC WSIs and in 40x magnification for H&E WSIs. Tiles with 40x magnification were created with a size of 60 x 60 μm, which corresponds to 237px x 237px. Tile sizes at 20x, 10x and 5x magnification are 120x120 μm, 240x240 μm and 480x480 μm, respectively. All tiles used for training, validation and hold-out testing were extracted without stride/overlap.

## Models

To classify pigmented lesions between melanomas and nevi, the ResNet architecture introduced by *He et al.* [22] was selected as a model for all data modalities. The hyperparameters of the different models were tuned individually using the Bayesian optimization framework Optuna [23] and five-fold cross-validation, all the models where load from the *timm* library [24] To avoid overfitting with respect to slides containing a huge amount of tiles, we used weighted sampling to train with a predefined amount of tiles per slide in all epochs. The hyperparameters we tuned were the size of the ResNet, the learning rate, the number of training epochs, the type of pooling, the number of tiles used per training epoch and whether or not the initialized ResNet was pretrained on ImageNet. The parameterization of all models is shown in the supplements in S1 Table.

The slide prediction procedure for the different image modalities is as follows: The models were trained at the tile level (with resolutions of 0.25, 0.5, 1.0 and 2.0μm/px), using the slide label for each tile of the slide. All tiles of the slide were predicted and the slide score was calculated by averaging all tile scores (Fig 1). To train models capable of handling domain shifts, the color jitter augmentation package of PyTorch [25] was used as part of the training process, as mentioned by *Tellez et al.* [26]. In contrast to H&E stained slides, features of protein expression can be distributed over a larger area in the cytoplasm. For this purpose, different magnifications were used to analyze these larger features.

## Combined models

Unimodal classifiers were combined to build models based on multiple data modalities. A classifier based on all four MelanA magnifications was built, where predictions with higher certainty give a higher contribution to the combined prediction. Scores of the different magnification models were averaged and weighted based on their distance to the optimal

**Table 1. Description of the population in our datasets.** For continuous features we report median, range, and number of NAs, for categorical features we report the total number of observations per group. Here the training population as well as all three test populations are described. Melanoma in situ describes the early stage of a malignant melanoma that has not yet broken through the basement membrane. However, features at the cellular level do not differ between melanoma in situ and malignant melanoma.

| | | Melanoma | Melanoma in situ | Nevi | All |
|---|---|---|---|---|---|
| | | | Dresden (train) | | |
| Samples | | 82 | 13 | 112 | 207 |
| Age | | 70[29;95] | 74[43;87] | 44[18;94] | 61 [18;95] |
| Breslow | | 0.7[0.0;20.0] 8 NA | 0.4[0.0;0.6] 6 NA | 112 NA | 0.6[0.0;20.0] 126 NA |
| Gender | male | 52 | 10 | 49 | 111 |
| | female | 30 | 3 | 63 | 96 |
| AJCC stage | 0 | 0 | 13 | 0 | 13 |
| | I | 60 | 0 | 0 | 60 |
| | II | 14 | 0 | 0 | 14 |
| | III | 8 | 0 | 0 | 8 |
| | NA | 0 | 0 | 112 | 112 |
| Localisation | Extremities | 27 | 1 | 34 | 62 |
| | Head | 7 | 6 | 12 | 25 |
| | Trunk | 48 | 6 | 66 | 120 |
| | | | Dresden (test) | | |
| | Samples | 45 | 15 | 66 | 126 |
| Age | | 73[33;92] | 69[43;92] | 59[20;88] | 67 [20;92] |
| Breslow | | 0.9[0.3;6.5] 5 NA | 0.0[0.0;0.3] 12 NA | 66 NA | 0.7[0.0;6.5] 83 NA |
| Gender | male | 24 | 6 | 31 | 61 |
| | female | 21 | 9 | 35 | 65 |
| AJCC stage | 0 | 0 | 15 | 0 | 15 |
| | I | 29 | 0 | 0 | 29 |
| | II | 10 | 0 | 0 | 10 |
| | III | 3 | 0 | 0 | 3 |
| | NA | 3 | 0 | 66 | 69 |
| Localisation | Extremities | 15 | 7 | 21 | 43 |
| | Head | 13 | 4 | 7 | 24 |
| | Trunk | 17 | 4 | 38 | 59 |
| | | | Erlangen | | |
| Samples | | 41 | 5 | 35 | 81 |
| Age | | 62[34;93] | 64[48;86] | 51[23;83] | 57[23;93] |
| Breslow | | 0.5[0.0;10.0] 3 NA | 0.5[0.5;0.5] 4 NA | 35 NA | 0.5[0.0;10.0] 39 NA |
| Gender | male | 22 | 3 | 23 | 48 |
| | female | 19 | 2 | 12 | 33 |
| AJCC stage | 0 | 0 | 5 | 0 | 5 |
| | I | 21 | 0 | 0 | 21 |
| | II | 7 | 0 | 0 | 7 |
| | III | 4 | 0 | 0 | 4 |
| | IV | 1 | 0 | 0 | 1 |
| | NA | 8 | 0 | 35 | 35 |
| Localisation | Extremities | 14 | 1 | 17 | 32 |
| | Head | 11 | 1 | 1 | 13 |
| | Trunk | 16 | 3 | 17 | 36 |
| | | | Naples | | |
| Samples | | 15 | 10 | 25 | 50 |

*(Continued)*

**Table 1.** (Continued)

| | | Melanoma | Melanoma in situ | Nevi | All |
|---|---|---|---|---|---|
| | | \multicolumn{4}{c|}{Dresden (train)} | | | |
| Age | | 51[28;71] | 66[51;84] | 35[21;80] | 49[21;84] |
| Breslow | | 2.0[0.4;5.8] | 10 NA | 25 NA | 2.0[0.4;5.8] 35 NA |
| Gender | male | 5 | 6 | 10 | 21 |
| | female | 10 | 4 | 15 | 29 |
| AJCC stadium | 0 | 0 | 10 | 0 | 10 |
| | I | 7 | 0 | 0 | 7 |
| | II | 0 | 0 | 0 | 0 |
| | III | 7 | 0 | 0 | 7 |
| | IV | 1 | 0 | 0 | 1 |
| | NA | 0 | 0 | 25 | 25 |
| Localisation | Extremities | 6 | 2 | 9 | 17 |
| | Head | 1 | 2 | 4 | 7 |
| | Trunk | 7 | 6 | 12 | 25 |
| | NA | 1 | 0 | 0 | 1 |

decision threshold. Other fusion approaches like averaging the scores unweighted or weighted based on the model's validation performances were investigated, all of which yielded comparable results (shown in the supplements, S2 Table). The H&E classifier was combined with the MelanA multiscale classifier using the same fusion method.

Motivated through the clinical practice we investigated another setup, called the hierarchical setup, where we first predict the label based on the H&E-classifier but add the MelanA based classifier for those lesions where the H&E WSI leads to an uncertain prediction only.

To calculate whether or not a H&E-based prediction was uncertain, we calculated confidence intervals (CIs) of the slide-level score via bootstrap and checked afterwards whether the optimal decision threshold is contained in the 95% CI. For cases where the threshold was contained in the CI of the slide-level score we added the MelanA based classifier.

## Reporting

For all results, 95% CIs are given next to the corresponding Areas under the receiver operating characteristic curve (AUROCs) of the model. CIs were calculated using the bootstrap-method [27]. The method was applied to the predicted values of a cohort. AUROCs were then calculated for this bootstrap cohort. After 10,000 repetitions the 2.5% as well as the 97.5% quantiles and thus, the 95% CI were calculated.

## Results

W The unimodal H&E classifier is based on previous works [5, 10, 28–30] and was adapted to the corresponding MelanA resolutions. The MelanA-models were combined and fused with

**Table 2. Antibodies and parameters of staining methods used by the different clinics.**

| Hospital | Clone | Company | Stain machine | Kit | Dilution |
|---|---|---|---|---|---|
| Dresden | A103 | Agilent | Ventana Roche Benchmark Ultra | ultraView Red | 1:25 |
| Erlangen | A103 | Millipore Sigma | Roche BenchMark XT | Fast Red | 1:200 |
| Naples | A103 | Ventana | Ventana Roche Benchmark Ultra | ultraView Red | 1:1 |

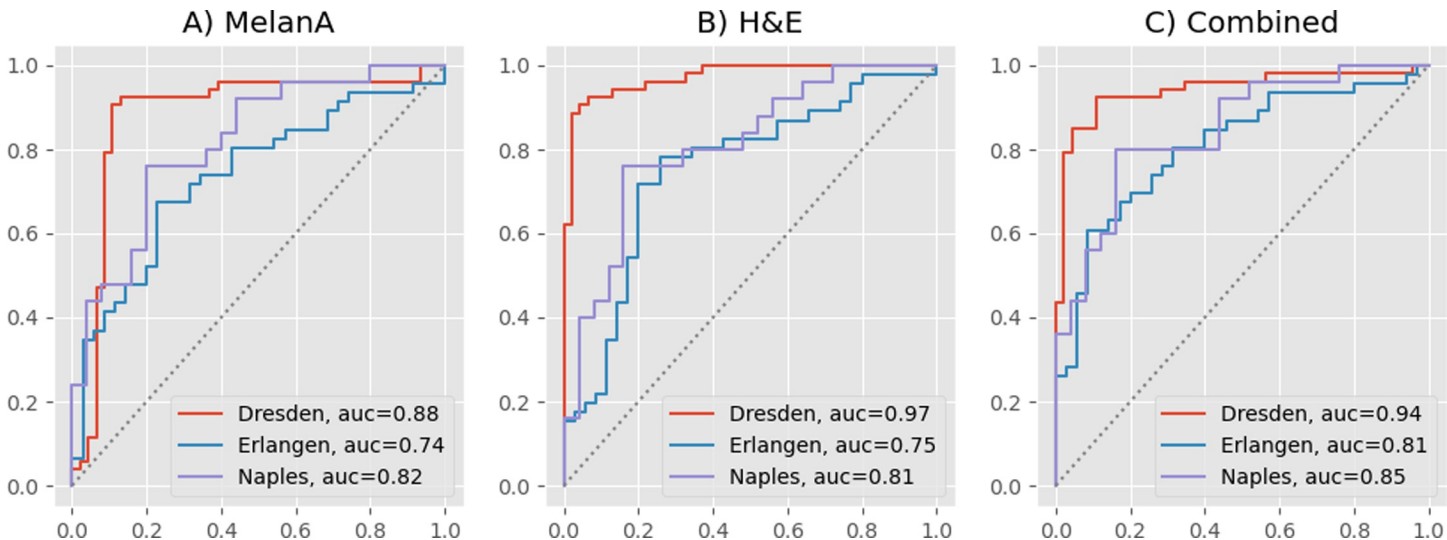

**Fig 1. Schematic diagram of the different models.** The red box shows the pipeline for MelanA-stained WSIs and the purple box the pipeline for H&E-stained WSIs. We tessellated MelanA-stained WSIs corresponding to different magnifications and trained individual models on each tile size. The class probabilities for each tile were predicted and aggregated into a slide score by averaging all tile scores. For the H&E-based model we proceeded in the same way.

the H&E model into one multi-modal classifier. All described models were tested within internal distribution (InD) on the Dresden holdout set, and OOD on the Erlangen and Naples cohorts (see Table 1). AUROCs and bootstrapped CIs for all models are shown in Table 3.

All results differ significantly from random guessing since no CI contains 0.5, the critical value. Thus, we are able to classify melanoma on all evolved MelanA-based models as well as with the benchmark H&E-based model as well. Beside this, it should be highlighted that almost all models on all cohorts perform with a AUROC significantly better than 0.7 which makes findings probably relevant for clinical practice. However, note that CIs overlap in several cases, indicating that different models perform similarly and thus, probably contain a high amount of shared Information.

In addition, we investigated another hierarchical approach motivated by clinical practice, using only MelanA-stained slides for cases where the H&E-based model is uncertain.

The ROC diagrams of the MelanA-based, the H&E-based, and the combined models for all three cohorts are shown in Fig 2. Another representation of this plot, to better compare models within one cohort is shown in the supplements in S4 Fig. Additional ROC plots of the individual MelanA models, which consider only one magnification, and results of the hierarchical approach are shown in the supplementary material in S2 and S3 Figs.

**Table 3. AUROC values as well as 95% bootstrapped CIs for the three test cohorts and all evolved models.**

| Stains and resolutions used | AUROC (Dresden) | AUROC (Erlangen) | AUROC (Naples) |
|---|---|---|---|
| H&E (0.25 µm/px) | 0.96 [0.94;0.99] | 0.75 [0.64;0.86] | 0.81 [0.67;0.92] |
| MelanA (0.25 µm/px) | 0.90 [0.84;0.97] | 0.75 [0.64;0.85] | 0.79 [0.66;0.91] |
| MelanA (0.50 µm/px) | 0.92 [0.87;0.97] | 0.78 [0.67;0.87] | 0.77 [0.63;0.89] |
| MelanA (1.00 µm/px) | 0.88 [0.82;0.95] | 0.73 [0.62;0.84] | 0.8 [0.67;0.92] |
| MelanA (2.00 µm/px) | 0.86 [0.78;0.95] | 0.67 [0.52;0.77] | 0.75 [0.60;0.87] |
| MelanA (all 4 combined) | 0.88 [0.80;0.96] | 0.74 [0.62;0.84] | 0.82 [0.68;0.92] |
| MelanA + H&E | 0.94 [0.89;0.98] | 0.81 [0.71;0.90] | 0.85 [0.73;0.94] |
| MelanA + H&E (hierarchical) | 0.96 [0.95;1.00] | 0.75 [0.64;0.86] | 0.83 [0.71;0.94] |

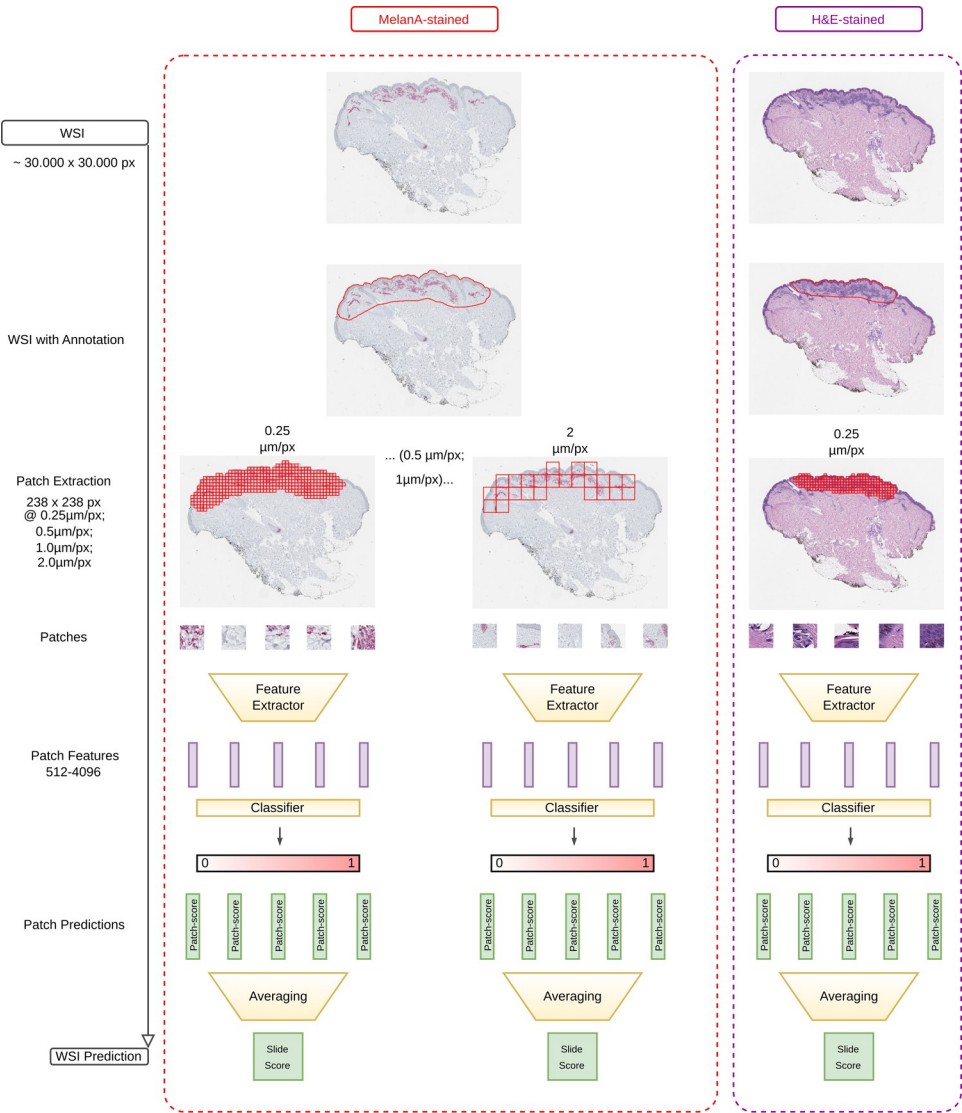

**Fig 2. ROC plots by data modality with corresponding AUROC values.** The different subplots show results for the individual evolved models: **A**: MelanA-based performance **B**: H&E-based performance taking all magnifications into account **C**: combined model using H&E as well as MelanA by aggregating the individual scores. The different colors of the ROC curves show from which data source site the results come: **Red**: internal results (Dresden), **Blue**: external results (Erlangen), **Purple**: external results (Naples).

## MelanA-based classifiers

The curves of the different magnifications are shown in the supplementary material in S1 Fig. They overlap at several points in all cohorts, which means that for different sensitivity/specificity trade-offs, different magnifications lead to the best results. In the internal cohort, the classifiers reached AUROCs between 0.85 and 0.92, in the Erlangen cohort AUROCs between 0.67 and 0.78, and in the Naples cohort AUROCs between 0.75 and 0.80. The CIs of the different magnifications overlap in all cohorts, so there is no magnification that leads to a significantly best performance overall.

The combination of all 4 magnifications, shown in Fig 2 A), was not significantly different from the models that use only one magnification.

In the Dresden (0.88) and Erlangen (0.74) cohorts, the AUROC of the combined MelanA model without considering CIs is worse than that of the (0.50 μm/px) model as a stand-alone classifier. For the Naples cohort, the AUROC of the combined MelanA classifier (0.82) is slightly, but not significantly, better than all individual models.

### H&E-based classifier

The classifier using only H&E-stained tissue, as our benchmark, achieved an AUROC of 0.96 on the internal test set and AUROCs of 0.75 and 0.81 on the external cohorts, respectively. The ROC plot in Fig 2 B) and the results in Table 3 show that the internal performance is significantly better than the external performance. Performance on both external data sets is not significantly different.

### Combined classifiers using H&E and MelanA

The model based on both data modalities, the H&E-stained tissue as well as the MelanA-stained tissue of all investigated resolutions, shown in Fig 2 C), performs numerically slightly worse compared to the H&E model on the Dresden cohort, reaching an AUROC of 0.94.

However, in the external cohorts the combined model performs best in absolute numbers, reaching AUROCs of 0.81 and 0.85 on the Erlangen and Naples cohorts, respectively. Nevertheless, the performance of the combined model is not significantly different from the MelanA-based model or from the H&E-based model for any of the investigated cohorts.

The hierarchical approach, where MelanA predictions are only taken into account when H&E-based prediction is uncertain, which reflects the diagnostic path better, leads to ROC-plots shown in S3 Fig. This approach resulted in the numerically best, albeit still not significantly different, performance on the internal cohort. It did not change results on Naples, the smaller external cohorts, since the H&E-based model was only uncertain for one sample within the Naples cohort and was certain for all samples in the Erlangen cohort.

## Discussion

In this work, we were able to predict melanoma/nevi classification across multiple datasets on MelanA slides with a similar accuracy as on benchmark H&E slides using DL-based image analysis. Furthermore, the results may suggest that the multistain approach has the potential to improve prediction accuracy and robustness, since at least on both external cohorts the combined model reached the highest AUROCs.

To integrate the presented work into clinical practice, a method for AI-pathologist interaction needs to be developed. For this purpose, we are developing an Explainable artificial intelligence system in collaboration with dermatologists [31], which produces easily interpretable explanations based on dermatoscopic images and aims to be integrated as an AI tool into digital pathology and clinical practice. Such a system can be expanded to include other data modalities such as immunohistochemistry or routine histology.

In clinical practice, pathologists often use H&E-stained tissue sections for melanoma diagnosis and resort to IHC-stained tissue in uncertain cases [32]. While DL-assisted detection of melanoma on H&E sections has been well studied [7], few studies have been performed using additional routine IHC-stained slides. Digital image analysis by automated quantification of the proliferation marker Ki-67 was used to distinguish melanoma from nevi as a diagnostic and prognostic aid [33]. Recently, an improved DL annotation method for H&E/SOX10 dual stains was developed to better identify tumor cells in cutaneous melanoma [34]. In the study presented here, MelanA-stained tissue was selected as an additional diagnostic tool since it highlights the cytomorphology and the distribution of melanocytes, thereby allowing a more

accurate evaluation of the architecture of any melanocytic tumor, along with the size and the shape of single cells. Other IHC stains such as HMB45, p16, and PRAME were excluded because they are useful only in selected cases. However, SOX10 was not chosen because it is a nuclear marker and gives no idea about the actual size of melanocytes and about the morphologic features of their dendritic processes. Finally, Ki67, although largely used in routine, is of little help in the recognition of in situ and early invasive melanoma [13–16].

In the current pathological routine, IHC markers including MelanA are used heterogeneously in different hospitals and laboratories. At the university hospital in Dresden, generally all dermatologically melanoma-suspicious skin lesions are stained with MelanA, providing an unbiased training dataset for our study. In contrast, the OOD datasets likely contain more challenging lesions since MelanA-stained tissue was only prepared at the university hospital in Erlangen in case the H&E-stained slides provided uncertain pathological results. The Naples dataset contained 40% in situ melanomas, all of which are small in size and generate few tiles, making classification in general potentially difficult.

The Dresden test set apparently does not benefit from the inclusion of additional data (S4 Fig), since the H&E-stained tissue slides are already sufficient to yield maximum accuracy. This may be due to the rather unambiguous dataset and thus, the very high performance and a broad data set with many subclasses. In contrast, the OOD datasets benefit from incorporating the additional MelanA-stained slides, making the classifier externally more robust. A combined classifier thus provides an advantage here, a finding we have already made in predicting BRAF status using H&E, clinical and methylation data in melanoma [10] suggesting that a multi stain based classifier can lead to better generalizability. Although the information contained in the H&E- and the MelanA-stained slides is probably partially redundant, one can still see a benefit of combining both stains on OOD data.

A detailed analysis of the various misclassifications revealed that it was not individual histopathological features that caused the model to underperform, but mainly technical artifacts such as overlapping section fragments, low staining intensity combined with strong pigmentation and very small lesions that resulted in only a small number of tiles. This also explains why the MelanA+H&E classifier did not perform better in the Dresden cohort. The simple H&E classifier is already sufficiently good; a combination in the Dresden hold out dataset does not lead to any improvement in individual cases, as there are individual unseen technical coloring artifacts here. This could be avoided by increasing the amount of data in the training set or by excluding the staining artifacts. In clinical pathology, MelanA staining is used in parallel as well as in addition to H&E staining to perform melanoma and nevi classifications. The combined model design was conceived as a parallel evaluation, whereby the model has the greatest possible information at its disposal.

Due to the cytoplasmic distribution of the MelanA protein, tiles from a higher magnification can potentially be too small to extract all relevant features. Pathologists frequently investigate the MelanA stains at lower magnifications to evaluate the silhouette and overall architecture of the lesion, which also contains valuable information. Our data could not show that there is an identifiable best magnification. However, each magnification seems to contain partly different information as the combination of all 4 magnifications brings a slight overall improvement, which can be attributed to ensembling.

Contrary to clinical practice, the hierarchical approach did not lead to any improvement on external datasets. This shows that an unbiased dataset is preferable for training a DL model, since the network can make better decisions with larger datasets. Interestingly, the uncertain Dresden specimens are lesions with large diameters of 8 mm to 17 mm, where a melanoma has developed in the center of a nevus, with melanoma features smoothly merge into nevus

features which probably confuses the model, as all tiles are weighted equally in our model. In contrast, the uncertain lesion in Naples is very small with a diameter of <1.0 mm.

## Limitations

Overall, the major limitation of this study is the relatively small sample size of the external test sets. In addition, the above-mentioned variability in the pathological routine as well as the different staining protocols of the respective clinics complicate the comparison of the results and findings. In addition, a not inconsiderable label noise must be taken into account, since the labels were histopathologically verified according to the gold standard of care, but a high inter-rater variability must be assumed, as shown in previous studies [3, 35].

## Conclusions

With DL analysis of MelanA-stained tissue, we were able to classify melanomas and nevi in two distinct OOD cohorts with similar accuracy as with H&E-stained tissue. The numerically, but not statistically significantly, better classification results achieved by combining H&E and MelanA classifiers suggests that the combination of these image modalities may lead to improved generalizability and performance. However, these results need to be confirmed in larger studies containing more lesions.

## Supporting information

**S1 Fig. Representative thumbnails for melanoma, melanoma in-situ, and nevus for all three cohorts.**
(TIF)

**S2 Fig. ROC plots by data source site with corresponding AUROC values. A**: Results from Dresden **B**: Results from Erlangen **C**: Results from Naples. **Red**: 40x magnification **Blue**: 20x magnification **Purple:** 10x magnification **Gray:** 5x magnification.
(TIF)

**S3 Fig. ROC plot of the hierarchical compared to the combined approach with corresponding AUROC values by data source site. A**: Results from Dresden **B**: Results from Erlangen **C**: Results from Naples. **Black:** Results of the combined approach using H&E and MElanA for all lesions **Red:** Hierarchical approach using MelanA-stained tissue only for H&E-based uncertain lesions.
(TIF)

**S4 Fig. ROC plots by data modality with corresponding AUROC values. A**: Results from Dresden **B**: Results from Erlangen **C**: Results from Naples. **Red**: MelanA-based performance taking all magnifications into account **Purple**: H&E-based performance **Black**: combined model using H&E as well as MelanA by aggregating the individual scores.
(TIF)

**S1 Table. Hyperparameters of all evolved models.**
(XLSX)

**S2 Table. Additional results derived by using different fusion approaches: Dist-opt means weighted by the distance to the individual models optimal thresholds; dist-05 means weighted by the distance to the default threshold of 0.5; avg denotes the fusion by conducting a simple average of all scores; perf means weighted based on the individual models validation performance in a way that better performing models contribute more to the fused**

**result.**
(XLSX)

**S1 Dataset.**
(ZIP)

**S2 Dataset.**
(ZIP)

**S3 Dataset.**
(ZIP)

**S4 Dataset.**
(ZIP)

**S5 Dataset.**
(ZIP)

## Author Contributions

**Conceptualization:** Christoph Wies, Lucas Schneider, Gerardo Ferrara, Eva I. Krieghoff-Henning, Titus J. Brinker.

**Data curation:** Christoph Wies, Lucas Schneider.

**Formal analysis:** Christoph Wies.

**Funding acquisition:** Eva I. Krieghoff-Henning, Titus J. Brinker.

**Investigation:** Christoph Wies, Lucas Schneider.

**Methodology:** Christoph Wies, Lucas Schneider.

**Project administration:** Lucas Schneider, Tabea-Clara Bucher, Eva I. Krieghoff-Henning, Titus J. Brinker.

**Resources:** Sarah Haggenmüller, Sarah Hobelsberger, Markus V. Heppt, Gerardo Ferrara.

**Software:** Christoph Wies.

**Supervision:** Lucas Schneider, Eva I. Krieghoff-Henning, Titus J. Brinker.

**Validation:** Lucas Schneider.

**Visualization:** Christoph Wies, Lucas Schneider.

**Writing – original draft:** Christoph Wies, Lucas Schneider.

**Writing – review & editing:** Sarah Haggenmüller, Tabea-Clara Bucher, Sarah Hobelsberger, Markus V. Heppt, Gerardo Ferrara, Eva I. Krieghoff-Henning, Titus J. Brinker.

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
