## [Decision Letter · Decision Letter 0]

16 Oct 2023

PONE-D-23-29059Evaluating Deep Learning-based Melanoma Classification using Immunohistochemistry and Routine Histology: A Three Center StudyPLOS ONE

Dear Dr. Brinker,

Thank you for submitting your manuscript to PLOS ONE. After careful consideration, we feel that it has merit but does not fully meet PLOS ONE’s publication criteria as it currently stands. Therefore, we invite you to submit a revised version of the manuscript that addresses the points raised during the review process.

We look forward to receiving your revised manuscript.

Kind regards,

Vincenzo L'Imperio, MD

Academic Editor

PLOS ONE

Journal Requirements:

2. Thank you for including your ethics statement:  "Ethics approval was obtained from the ethics committees of the

respective universities before the study was initiated. Patients provided informed written consent. This work was

performed in accordance with the Declaration of Helsinki". 

[TJB would like to disclose that he is the owner of Smart Health Heidelberg GmbH (Handschuhsheimer Landstr. 9/1, 69120 Heidelberg, Germany) which develops mobile apps, outside of the submitted work. SHo reports clinical trial support from Almirall and speaker’s honoraria from Almirall, UCB and AbbVie and has received travel support from the following companies: UCB, Janssen Cilag, Almirall, Novartis, Lilly, LEO Pharma and AbbVie outside the submitted work.]. 

7. We note that you have included the phrase “data not shown” in your manuscript. Unfortunately, this does not meet our data sharing requirements. PLOS does not permit references to inaccessible data. We require that authors provide all relevant data within the paper, Supporting Information files, or in an acceptable, public repository. Please add a citation to support this phrase or upload the data that corresponds with these findings to a stable repository (such as Figshare or Dryad) and provide and URLs, DOIs, or accession numbers that may be used to access these data. Or, if the data are not a core part of the research being presented in your study, we ask that you remove the phrase that refers to these data.

8. Your ethics statement should only appear in the Methods section of your manuscript. If your ethics statement is written in any section besides the Methods, please move it to the Methods section and delete it from any other section. Please ensure that your ethics statement is included in your manuscript, as the ethics statement entered into the online submission form will not be published alongside your manuscript. 

9. We notice that your supplementary figures and tables are included in the manuscript file. Please remove them and upload them with the file type 'Supporting Information'. Please ensure that each Supporting Information file has a legend listed in the manuscript after the references list.

10. Please include captions for your Supporting Information files at the end of your manuscript, and update any in-text citations to match accordingly. Please see our Supporting Information guidelines for more information: http://journals.plos.org/plosone/s/supporting-information. 

11. We note that Figure 1 in your submission contain copyrighted images. All PLOS content is published under the Creative Commons Attribution License (CC BY 4.0), which means that the manuscript, images, and Supporting Information files will be freely available online, and any third party is permitted to access, download, copy, distribute, and use these materials in any way, even commercially, with proper attribution. For more information, see our copyright guidelines: http://journals.plos.org/plosone/s/licenses-and-copyright.

A. You may seek permission from the original copyright holder of Figure 1 to publish the content specifically under the CC BY 4.0 license. 

B. If you are unable to obtain permission from the original copyright holder to publish these figures under the CC BY 4.0 license or if the copyright holder’s requirements are incompatible with the CC BY 4.0 license, please either i) remove the figure or ii) supply a replacement figure that complies with the CC BY 4.0 license. Please check copyright information on all replacement figures and update the figure caption with source information. If applicable, please specify in the figure caption text when a figure is similar but not identical to the original image and is therefore for illustrative purposes only.

Additional Editor Comments (if provided):

Reviewers' comments:

Reviewer's Responses to Questions

**Comments to the Author**

1. Is the manuscript technically sound, and do the data support the conclusions?

Reviewer #1: Yes

Reviewer #2: Yes

2. Has the statistical analysis been performed appropriately and rigorously? 

Reviewer #1: Yes

Reviewer #2: Yes

3. Have the authors made all data underlying the findings in their manuscript fully available?

Reviewer #1: Yes

Reviewer #2: No

4. Is the manuscript presented in an intelligible fashion and written in standard English?

Reviewer #1: Yes

Reviewer #2: Yes

5. Review Comments to the Author

Reviewer #1: Titus J. Brinker et al.’s work presents an original computational approach to melanocytic lesions, utilizing clear and elegant methods, yielding promising yet not optimal results for clinical translation. Specifically, the study employs Deep-learning models to analyze histological information extracted from both H&E and Melan-A immunostain, aiming to classify lesions into benign nevus or melanoma. The study is well-executed and well-designed, however, as mentioned in the paper, this approach has certain limitations. One of its inherent shortcomings is the inability to provide a comprehensive evaluation of the entire lesion at low magnification, which is usually crucial in the evaluation of melanocytic lesions, overlooking important features such as the symmetry. Another limitation is the lack of a direct link to histological characteristics and the absence of accompanying histological images, which could significantly improve the accessibility and clarity of the study. On the other hand, the study's main strengths include the evaluation of different immunohistochemical clones, a multicentric approach with a well-balanced dataset and training conducted at various magnifications.

Overall, the study could be improved with a few adjustments:

-In the introduction, it is mentioned that pathologists have up to a 25% interobserver discordance rate, but the references cited are relatively dated (from 15 to 27 years ago), a period during which the evaluation of melanocytic lesions has gained new markers and insights. More recent references would be advisable.

-In the same section, from the sentence beginning with 'If melanoma is initially misclassified...' to the end of the paragraph, references are lacking.

-In the Methods section, Table 1 categorizes cases into three classes, including melanoma in situ. While it is implied that melanoma in situ belongs to the melanoma class, the table's presentation may still cause confusion. To enhance clarity, it is advisable to explicitly specify the relationship between melanoma in situ and the melanoma class.

-In the same section it is stated that a color augmentation approach was used, but it is not established whether this led to improvements in metrics and domain shift. At least some commentary on this approach, which sparks debate on model cutoff changes, would be recommended.

-In the results section, the initial part appears somewhat repetitive, and it would be advisable to condense this portion within the methods to make the entire methodological process more straightforward.

-Still in the results section, the legend in Figure 2 appears to reverse Figures 2A and 2B.

-In the discussion, no explanation or hypothesis is provided for these misclassifications, and it is not described from a histological perspective in which cases the model underperformed. It remains unclear whether this underperformance was solely due to variations in technical slide characteristics or if there are specific subsets of lesions where the model struggled more.

Reviewer #2: Since the objective of the study is not simply the development of a DL-based model to diagnose melanoma on WSI, but a comparison of the performance of DL systems using H&E versus MelanA, without lengthening the introduction, I would integrate it by briefly mentioning a few numbers, namely: (1a) the current state-of-the-art performance of published DL models in melanoma diagnosis, (1b) whether there are public datasets to gauge performance, and (2) the reported accuracy boosts by combining H&E with IHC (in other use-cases if there are none in melanoma).

It would be useful for the reader if the authors provided a few example WSI thumbnails and fields from the three cohorts, for each staining modality, to gauge the entity of staining/scanning variation between cohorts.

Am I correct in understanding that the patches were extracted without stride/overlap? Was the same done for inference? I would state this clearly in the manuscript.

Some details regarding model architecture are lacking. In addition to the mentioned pooling strategy, where were models cut, and what custom heads were used?

What hardware was used? What's the inference time per tile and per WSI?

L127. Training the model at tile level using the slide label has some risks. Were there WSIs in which multiple categories coexisted (i.e. melanoma ± in-situ melanoma ± benign nevus)? How were these handled in training? And in inference?

The model input is (a batch of) 237×237px patches, but what about its output? Figure 1 seems to show a binary output (melanoma vs benign?) but you also had in-situ melanoma in the slide categories. What exactly was the output of the model?

L127 "All tiles of the slide were predicted and the slide score was calculated by averaging all tile scores (see Figure 1)". Can you provide more detail? The answer to this question depends on the previous one. Was the model a binary melanoma/benign model, and was your final label "benign" if benign tiles outnumbered malignant tiles? Was your model a multicategory one (melanoma - melanoma in situ - benign nevus) and did you predict the most common category as the slide label?

How did you handle tiles which were not melanoma, melanoma in-situ or nevus? In the training set you annotated and extracted only tiles from the lesion, but what about the test set? Did you run inference on the whole tissue, including epidermis, dermis and subcutis uninvolved by melanocytic tumor? Did you use a tissue detector to filter out empty tiles?

The reported performances of the H&E-only and MelanA-only models make me think. The pathologists among the authors of this work are invited to discuss their interpretation of the fact that the MelanA+H&E classifier does not seem to work better than the simple H&E classifier, at least on the Dresden cohort. What does the pathologist's thought process look like when evaluating MelanA in addition to H&E, and was the combined model designed coherently?

L288 while I agree that each magnification contains partly different information, I don't think you can infer that from the observation that the combination of all 4 magnifications brings a slight overall improvement. Unless you prove that it is not due to ensembling.

6. PLOS authors have the option to publish the peer review history of their article (what does this mean?). If published, this will include your full peer review and any attached files.

Reviewer #1: **Yes: **Giorgio Cazzaniga

Reviewer #2: **Yes: **Alessandro Caputo

---

## [Author Response · Author response to Decision Letter 0]

22 Nov 2023

Dear Dr. L'Imperio, dear Dr. Cazzaniga, dear Dr. Caputo,

we thank the reviewers for their constructive suggestions and questions, which contributed to improving the quality of this work.

We attached an additional document "Review-Letter_with_comments" into the submission, where we respond point by point to all questions, the journal requirements as well as the specific questions from the two reviewers. 

We hope, that we could address all open points.

Sincerly,

Dr. Titus J. Brinker

---

## [Decision Letter · Decision Letter 1]

29 Dec 2023

Evaluating Deep Learning-based Melanoma Classification using Immunohistochemistry and Routine Histology: A Three Center Study

PONE-D-23-29059R1

Dear Dr. Brinker,

We’re pleased to inform you that your manuscript has been judged scientifically suitable for publication and will be formally accepted for publication once it meets all outstanding technical requirements.

Kind regards,

Vincenzo L'Imperio, MD

Academic Editor

PLOS ONE

Reviewers' comments:

Reviewer's Responses to Questions

**Comments to the Author**

1. If the authors have adequately addressed your comments raised in a previous round of review and you feel that this manuscript is now acceptable for publication, you may indicate that here to bypass the “Comments to the Author” section, enter your conflict of interest statement in the “Confidential to Editor” section, and submit your "Accept" recommendation.

Reviewer #1: All comments have been addressed

Reviewer #2: All comments have been addressed

2. Is the manuscript technically sound, and do the data support the conclusions?

Reviewer #1: Yes

Reviewer #2: Yes

3. Has the statistical analysis been performed appropriately and rigorously? 

Reviewer #1: Yes

Reviewer #2: Yes

4. Have the authors made all data underlying the findings in their manuscript fully available?

Reviewer #1: Yes

Reviewer #2: Yes

5. Is the manuscript presented in an intelligible fashion and written in standard English?

Reviewer #1: Yes

Reviewer #2: Yes

6. Review Comments to the Author

Reviewer #1: From my side, the authors have aptly addressed the comments. The refined manuscript is now well-prepared for publication.

Reviewer #2: All comments have been addressed adequately. The limitations that remain are explained in the manuscript text.

7. PLOS authors have the option to publish the peer review history of their article (what does this mean?). If published, this will include your full peer review and any attached files.

Reviewer #1: **Yes: **Giorgio Cazzaniga

Reviewer #2: **Yes: **Alessandro Caputo

---

## [Editor Report · Acceptance letter]

9 Jan 2024

PONE-D-23-29059R1 

PLOS ONE

Dear Dr. Brinker, 

I'm pleased to inform you that your manuscript has been deemed suitable for publication in PLOS ONE. Congratulations! Your manuscript is now being handed over to our production team.

Kind regards, 

on behalf of

Dr. Vincenzo L'Imperio 

Academic Editor

PLOS ONE